# Experimental Research into Conscious Self-Regulation in First-Graders with Developmental Delay

**DOI:** 10.3390/bs9120158

**Published:** 2019-12-15

**Authors:** Nataliya V. Babkina

**Affiliations:** Research Department, Institute of Special Education of Russian Academy of Education, Pogodinskaja str. 8/1, 119121 Moscow, Russia; natali.babkina@mail.ru; Tel.: +7-916-454-3890

**Keywords:** conscious self-regulation, children with developmental delay (DD), studying problems, inclusive education, levels of self-regulation development

## Abstract

The underdeveloped conscious self-regulation of actions plays a primary role in the disorder-related character of children with developmental delay (DD). This study aimed to analyze and systematize specific occurrences of conscious self-regulation in children’s DD-affected cognitive activity. Children aged 7–8 years were involved in the study (n = 60), including children with DD (n = 40) and school children displaying typical development (n = 20). The dotting test, visual pattern test, and Toulouse–Pieron attention test are the practical methods used in the study. Mathematical methods of statistics were applied to analyze the study outcome. The experimental research identified and described four levels of conscious self-regulation development in the cognitive activity of children with different learning capacities. The findings proved that children of 7–8 years with DD have an underdeveloped level of conscious self-regulation of actions in terms of their cognitive activity, and that they differ from their peers regarding typical development in terms of their conscious self-regulation level and skills. Special intervention programs that develop conscious self-regulation in the performance and behavior of children with DD are required to realize their cognitive and personal capacities and provide them with high-quality education.

## 1. Introduction

According to the holistic conceptual framework of Konopkin O.A. [1], the conscious self-regulation of actions is a system-like mental process of the initiation, development, and self-management of one’s conscious goal-guided activity, and it has certain types and forms.

Research into the common patterns of conscious self-regulation has understandably led to studies on its characteristic peculiarities (varying among different groups of children) and its specific occurrences in different types of activities. A distinct future research direction is conscious self-regulation in learning and cognitive activity. For instance, researchers have found correlations between learning motivation, levels of self-regulation development, and self-regulation styles [2,3,4,5,6,7,8]. School children with higher motivation (as opposed to their less-motivated peers) and with better self-regulation development proved to have better studying performance, a more positive attitude to learning, and a better capacity to adapt to changes in the educational process [9,10,11,12].

Furthering this research direction, we aim to study the peculiarities of conscious self-regulation in cognitive activity in children with mental developmental delay (DD), which is the most numerous and diverse subcategory of children with delays, who mostly take part in mainstream education classes and attend mainstream schools [13,14]. This will help to uncover children’s capacities and discover the conditions that reduce dysfunction to a minimum and/or overcome dysfunction.

The diversity of the etiological factors causing this type of mental dysontogenesis results in extreme heterogeneity between groups of children with DD. Factors such as the organic and/or functional impairment of the central nervous system, constitutional factors, chronic somatic diseases, poor upbringing conditions, and mental and social deprivation can be found among the causal roots of developmental delay. As a result, the manifestation of the delay has a significant range: from conditions such as age norms to cases that need to be differentiated from mental retardation. This is why the capacities of children with DD and the special-needs conditions that facilitate compensation for developmental delay will differ from child to child, and a differentiated approach is required to provide intervention services.

According to psychological and educational studies, it is characteristic of children with DD that they are impulsive and unable to reach their goals when performing tasks, that their mental action-planning skills and self-control are underdeveloped, that they have difficulty maintaining interest in tasks, and that they exhibit poor levels of new learning [13,14,15,16].

Studies on the emotional component of self-regulation [15] proved that children with DD are emotionally immature and unstable, which is manifest in their impulsive and rude behavior, aggression, elevated social classroom problems, and irritability, meaning that they are not able to behave and act in compliance with conventional rules. This all makes their adaptation to school educational norms and conditions more difficult. However, these studies show that timely intervention services can have a positive impact on the cognitive and personal development of children with DD. Even in non-mainstream schools (classes) for children with special needs, children’s self-regulation skills remain underdeveloped by the end of their school education [17,18] when intervention programs do not adequately target the development of self-regulation in children with DD. This can, for instance, cause them to have severe socializing problems [19].

It is evident that the nature of self-regulation difficulties in children with DD is complex, as it deals with many factors, including cortical neurodynamic disorders, aspects of the emotional sphere, willpower, and parenting gaps at previous stages of development.

In this research, self-regulation is considered to be a special integrative characteristic (as part of a system) of children’s mental activity. It can be revealed through psychological experiments and is therefore accessible for direct observation data collection, monitoring, description, and classification, according to the levels and possible variants of its development. 

This study aimed to profoundly research a system-like process of conscious self-regulation in first-graders with DD in terms of its peculiarities, structure, and content (self-regulatory skills). This is crucial for outlining the right conditions for self-regulation-focused intervention in order to achieve a targeted development of the self-regulatory sphere in children with DD.

## 2. Methods

Sixty children aged 7–8 years were involved in the study, with 20 schoolchildren with typical development and 40 with DD. The diagnostic screening of children with DD was performed by a panel of experts, comprised of a developmental pediatrician, an educational psychologist, and a special education teacher. 

Eight general schools in Moscow, Russia were selected for the sample. All the children were first-graders, and the children with DD studied in these schools according to the terms of inclusive education. This research was carried out with permission from the children’s parents (legal representatives), in compliance with Russian law.

Various conscious self-regulatory skills were studied, e.g., understanding activities, determining and adhering to an activity program, acquiring and maintaining the learned actions to use the techniques associated with a task, self-monitoring in the middle of a task, and making a final self-assessment by checking the final results. The analysis of the study’s outcome emphasizes the following efficacy indicators of self-regulation: work capacity fluctuations, skill automaticity readiness, and a transfer of self-regulation skills to new conditions. Different types of support and their impacts on the efficacy of self-regulation were also factored into the study. 

The dotting test, Toulouse-Pieron attention test, and visual pattern test are the traditional school readiness assessment methods applied in the research [20,21].

In the dotting test, children are asked to make dots on a test form using pencils and follow the experimenter’s cues (from simple to complex). First, children have a simple instruction to follow: to make as many dots as they can in a given time. Then, with the tasks gradually becoming more complex, children are asked to locate dots in groups. The test analyzes the efficiency of task performance, individual pace, and consistency in task performance. Within the testing method, there are four levels of complexity for the interpretation of the results. 

In the Toulouse-Pieron attention test, children are asked to match the test form figures according to the model symbols. Their performance is evaluated by the time needed to complete the task and by the number of mistakes.

In the visual pattern test, children are asked to reproduce a model pattern, introduced by an experimenter on a blackboard, and to continue the pattern sequence all the way to the end of a line. The accuracy of replicating the pattern and correctly continuing the sequence of the pattern are evaluated in the test.

In the research, we added some modifications to the protocol of the testing methods and expanded the number of measurable quality indicators of a child’s performance.

All the methods were applied to groups, not to individuals, in order to involve the self-regulatory capacities of the schoolchildren as much as possible and simulate real school conditions as the task conditions. Two groups were assembled: one consisting of children with DD, and another consisting of children with typical development. The number of schoolchildren simultaneously performing a task was 5 ± 1, allowing the experimenter to attentively watch every child’s record and report on all his/her actions, according to the protocols. 

The evaluation of screening results was based on a qualitative analysis of a child’s psychic activity and its peculiarities. The evaluated qualitative indicators for assessing a child’s performance in each task are listed below:-The child’s ability to organize his/her own activity: the way she/he starts to do the task, how the pre-task orienting is implemented (the pre-task orienting stage), how the performing process takes place (whether the actions are planned in an orderly manner and executed or are chaotic; whether impulsive behavior or “field-theory” behavior is characteristic of the participants or not);-The way a child choses to perform a task (rational/irrational); -The child’s ability to control and monitor his/her activity during task performance, notice mistakes, and find and correct them;-The child’s ability to use a model as a guideline: to complete a task by replicating or following a model, to compare his/her actions with the guidelines, and to manage step-by-step action control;-The child’s attitude to her/his own work results: whether the child shows interest in the final results, whether he/she exhibits an indifferent attitude, and whether the child is driven by the experimenter’s evaluation or by the objective results;-The child’s understanding of the task, responsiveness to support, and ability to transfer a demonstrated technique from one task to a similar type of task.

In order to study children’s readiness to learn how to organize their activity in children with DD, certain steps were designed to provide support in terms of stimulating and organizing the children’s activity. Accordingly, the external regulation of children’s actions was gradually increased, with support being provided to the children within the research. The amount of support that happened to be sufficient for a child to successfully complete a task was an indicator of the zone of proximal development [22], i.e., the capacities of a child that are activated when he/she is completing a task together with an adult. A full understanding of a task and what a child was asked to do, focusing on the task goal all the way through to task performance, step-by-step activity planning and management, result monitoring, checking work, and the final evaluation—all these characteristics of a child’s activity were considered in the analysis of the results. 

All these qualitative indicators were recorded during children’s task performance and were reported according to the protocols of the methods used in the subsequent analysis.

When possible, the quantitative indicators of a child performance (time needed to complete a task, number of mistakes, etc.) were also subject to analysis. Mathematical methods of statistics were applied to analyze the study outcome.

## 3. Results

### 3.1. Conscious Self-Regulation Study

The dotting test, modified by Osnitsky A.K. [20], was applied to study the self-regulation of actions and their capacity (determining and adhering to an activity, acquiring and maintaining the learned actions needed to perform tasks, and learning self-regulation skills that are transferrable to new conditions), as well as the impact of different conditions on the performance and efficacy of self-regulation in children with DD as opposed to their peers with typical development.

The conscious self-regulation of actions, individual pace, and mental performance capacity were factored in when processing the outcome data. 

Four levels of the conscious self-regulation of actions were outlined in order to execute qualitative analysis and data interpretation. On the first level are the self-regulation skills of actions in new conditions; on the second level are the self-regulation skills that are transferrable to new contexts (performing well and gaining good results by changing difficult conditions into easier ones, when possible); on the third level is a child’s ability to self-organize in familiar situations; and on the fourth level is the underdeveloped self-regulation of actions, even in situations that the child has experienced before.

The difference between the level of conscious self-regulation of actions in children with DD and in their peers with typical development was significant (*p* < 0.01; the p-value hereinafter is given by student’s T-test distribution). As shown in Figure 1, the majority of children with DD exhibited sufficient self-regulation development of actions in repeated situations (the third level: 65%). However, only 20% of the first-graders managed to transfer their self-regulation skills to slightly new contexts by themselves (the second level). An inability to repeat learned actions (the fourth level) was found in 15% of children with DD. There was no child with DD who exhibited the self-regulation of actions in new or unfamiliar contexts (the first level). Moreover, the majority showed an underdeveloped stability in task performance and difficulties in maintaining the efforts necessary to execute long-term tasks.

### 3.2. Self-Regulation Study

The Toulouse-Pieron attention test was applied in order to study the conscious self-regulation of activity (a child’s mental programming and control of his/her actions, adherence to instructions, and feature-specific attention allocation) and impacts of different types of support on self-regulation efficacy. 

In line with the research aim, two variants of tasks [19] were applied in the study. The first variant (the standard one) helped to analyze the activities in terms of goal-focused behavior (adherence to instructions, time-planning, tracking the dynamics of change in performance pace, and counting the mistakes). The second variant helped to analyze the children’s self-regulation capacity with the experimenter’s support in activity organizing (repeatedly reminding a child to be very attentive, not to rush, to focus on instructions, to do the task correctly, to do his/her best, and to check his/her task performance in terms of the task techniques). 

When a child began to perform a task from memory, the frequency with which he/she checks his/her work with the task sample, his/her verbal self-regulation of the activities, his/her attention distraction while performing the task, etc., were noted by the experimenter in the screening protocol.

The summarized results of the Toulouse-Pieron attention test (Figure 2) showed that the children with DD made significantly more mistakes, as compared to their peers with typical development (averages of 5.3 and 1.4, respectively, which is a significantly difference, with a p-value of *p* < 0.01), when a standard variant of the method (variant 1) was applied. On average, it took them more time to complete the tasks than the children with typical development (4′45″ and 3′20″, respectively).

The children with DD were lacking in regular and goal-oriented behavior in task performance. At times, they checked their own previous references, instead of checking with the task sample, which led to them repeating their mistakes. The children with DD rarely exhibited verbal self-regulation of activities.

The experimenter’s support in performance organizing (variant 2) provided to the children with DD did not result in a significant efficacy increase (3,9 and 4,5 mistakes, respectively, which is a non-significant difference; *p* < 0.05), whereas the children with typical development significantly improved their results (1.4 and 0.3; *p* < 0.05). It is explicitly shown that children with DD had difficulty in sequential task performance and checking results, though they understood the instructions well.

### 3.3. Conscious Self-Regulation Skills Study

The visual pattern test [20] was applied to study conscious self-regulation skills (performing the task and checking with the task sample, determining and adhering to an activity program, acquiring and maintaining the learned actions and task techniques, self-monitoring in the middle of a task, and making a final self-assessment by checking their final results and correcting the mistakes they had made).

Task precision in copying sample visual patterns and accuracy in using learned techniques in further copying of patterns (repeatedly drawing a pattern) were evaluated during the children’s task execution. Four levels of performance efficacy were outlined for further data analysis and interpretation. On the first efficacy-level, the children fully succeeded at performing the task and made no mistakes. On the second level, the children were also successful in performing the task but made a few mistakes that they corrected by themselves by checking their results with the sample pattern. On the third level, the children made no mistakes in copying the sample pattern but had difficulties in further copying the pattern (i.e., difficulties in adhering to an activity program and employing the task technique accurately without support). On the fourth level, the children could not manage to copy the sample visual pattern without support.

The test results (Figure 3) showed a significant difference between the children with DD and their peers with typical development (*p* < 0.01). The third level of efficacy turned out to be the most typical for the children with DD (60%). They made such mistakes as missing or changing some elements of the pattern and drawing the pattern beyond the grids on squared-notebook sheets (which was against instructions). Such mistakes occurred most frequently towards the end of a task. It was difficult for the children to focus on the goal. After the task was completed, they did not check what they had done with the sample pattern, and therefore did not see their mistakes. 

Some of the children with DD encountered difficulties from the very beginning when copying the pattern (the fourth level, 25%). Nonetheless, these first-graders managed to complete the task to different extents when the experiment conditions were changed (to individual tutoring).

## 4. Discussion

The generalized data analysis proved that 7- and 8-year-olds with DD have underdeveloped self-regulation of conscious actions in their cognitive and learning activity, and there is a statistically significant difference between children with DD and their peers with typical development in terms of the level of self-regulation in skills development. In the groups, for extended periods of time, it was difficult for the first-graders with DD to organize their efforts, to choose task techniques and sequentially perform tasks, to self-monitor in the middle of a task, and to make a final self-assessment by checking their final results and correcting their mistakes. As soon as the experiment conditions were adjusted, and the children were supported in activity organization (repeating instructions in a more simple way, providing emotional support, switching from a group activity to individual tuition, etc.), the children significantly improved their results.

The experimental study helped to outline four levels of the conscious self-regulation development of cognitive activity in children with different learning capacities (from “the high norm”—the first level—to pronounced mental developmental delay—the forth level).

In the first level, during the screening, the children do their best to make an effort, showing high performance and good focus level. Mental processing is developed at a high efficacy level. The children are capable of self-monitoring in the middle of the task and making a final self-assessment by checking their final results with the task sample by themselves. The task-goal adherence is maintained throughout the performance of a task, and the result assessment is valid. The following stages of the self-regulation of actions are clearly observed in children’s performance: pre-task orientation (goal setting and goal-guided behavior, identification of the key conditions), prediction (mental planning of actions), and performance (learning new actions and task techniques and maintaining them). An adult’s presence, extrinsic motivation, and the types of task introduced do not make a significant difference for these children. They are not situation-oriented. This level of self-regulation is typical mainly in children with a considerably high developmental norm.

In the second level, children engage in tasks eagerly but oftentimes impulsively, without proper pre-task orientation. After a while they are already prone to distraction from the performance of the task, which makes them need extrinsic motivation and self-actualization through personal experience. They are able to mentally plan their actions ahead in performing the task, to opt for task techniques and to sequentially execute them. Nevertheless, they are not able to self-monitor in the middle of the task and make a final self-assessment by checking their final results, unless an adult reminds them to do so. Mental processing development matches the age developmental milestones but is impeded without an adult’s support, which becomes obvious in the case of an adult’s non-presence or in frontal learning. Thus, children face difficulties and need extrinsic motivation at the pre-task orientation stage of conscious self-regulation, but the children are sufficiently developed in the orientation and prediction stages (they are able to plan their performance ahead, to opt for task techniques, and to sequentially execute tasks). The conscious self-regulation of cognitive activity (and its occurrences) in these children is situation-oriented. The effect of an adult’s presence and of a child’s personal experience on their performance efficacy is significant. This level of self-regulation is typical for children with mild DD.

In the third level, children engage in tasks eagerly, if the latter are introduced through games, and they only follow the overall goal of the activity and do not follow (or miss) most of the task rules. Their enduring focus on one type of activity and long-term efforts are both impeded. Normally, they need visual support tools when performing intellectual tasks. Their productive performance is usually provided not just by an adult’s presence but also by his/her active support at all stages: goal-setting and goal-guided performance, identification of the conditions needed to reach a goal, activity program setting and task technique choice, result checking, and mistake correction. These children’s critical thinking skills are underdeveloped, such that they cannot properly evaluate the results of their task performance (in most cases, they say that they completed the task without mistakes). The engaging but not completed tasks are evaluated by the children as easy ones. After one easy task is not correctly completed, the children ask for a more difficult one. Thus, these children encounter difficulties in both the pre-task orientation stage and at the prediction stage of self-regulation, while the performance stage is well developed. Being considerably situation-oriented, their success in task performance depends on an adult’s presence and on a form of task introduction. This level of self-regulation is characteristic of children with mild DD.

In the forth level, the children are reluctant to engage in the task, highly distractible, easily get tired, avoid intensive intellectual efforts, miss the point of the task, and focus on non-essential details. They exhibit a low level of mental performance and an inability to complete even the potentially approachable tasks by themselves. Therefore, an active adult’s support is needed not only during all the steps toward the task goal and the setting of conditions but also when performing actions. Oftentimes, introducing the task through games distracts children from the main point of the task, as he/she becomes more concerned with the game. The children’s self-evaluation and self-checking skills are, in fact, underdeveloped. Thus, these children encounter difficulties in all of the stages of the self-regulation of actions: in the pre-task orientation, prediction, and performance stages. They do not exhibit an explicit situation-dependency: they are equally unsuccessful at completing the task in both situations, with different extrinsic motivations and when exposed to different forms of task introduction. This level of self-regulation is mainly characteristic of children with severe DD.

The revealed peculiarities of conscious self-regulation in the cognitive activity of first-graders with DD explain why the children have difficulties that are described by the teachers as follows: school and learning skill issues, school rules and discipline problems, and peer problems, which lead to overall school maladjustment.

According to L. Vygotsky, self-regulation is an ability to manage oneself (one’s own external and internal actions) based on cultural predispositions. Here, self-regulation development can be considered as an acquirement of behavioral tools that can help a child to realize and control his/her own actions in order to build a certain behavior. As opposed to non-conscious self-regulation, conscious self-regulation has to do with socializing. Therefore, not only does self-regulation not depend so strongly on psychophysical prerequisites for regulation processes, it also compensates for them [22].

The conducted experimental research helped to outline the zone of actual and proximal development of conscious self-regulation in children with DD. According to L. Vygotsky, the zone of proximal development can be determined by the tasks that a child is yet to learn how to perform by him/herself but is already able to execute with an adult’s support. The skills and knowledge that are most approachable for a child only with an adult’s guidance then become skills and knowledge of his/her own. 

The study shows that children with DD have special needs that need to be met in order to develop conscious self-regulation: -Inclusion of self-regulation issues in the teaching and learning process;-Provision of special help in terms of children’s realization and overcoming self-regulation problems;-Work targeting the development of skills to organize activities, realize difficulties, ask for an adult’s support, and accept help;-Implementation of special tools of psychology and social and teaching sciences to activate children to independently deal with their every-day and learning tasks; -Special work on teaching children to transfer the skills gained through learning into their everyday life.

Well-organized exterior (first-hand) actions, based on the strategy of following a template and gradual interiorization of action planning (a plan to manage self-regulation of actions), should be the main way to develop the conscious self-regulation of cognitive activity. The method is based on the view of L. Vygotsky, reporting on the zone of proximal development and the crucial role of an adult in terms of the mental development of a child [22].

## 5. Conclusions

First-graders with DD exhibit underdeveloped conscious self-regulatory skills, which significantly impedes their adaptation to the conditions and requirements of school education. 

Special intervention services, focused on the development of the conscious self-regulation of actions and behavior, are required in order to realize cognitive and personal capacities in children with DD and for them to have access to good-quality education [23]. Such intervention services should be based on a differentiated approach and provided according to the individually diagnosed level of conscious self-regulation development in a child.

On the one hand, special conditions required to realize and unleash schoolchildren’s capacities should include a correct learning environment, designed to develop their major activities and build life skills and competencies relevant to their age. On the other hand, a special intervention aimed at the development of self-regulation skills should be made, and children should be encouraged to make efforts in this direction.

## Figures and Tables

**Figure 1 behavsci-09-00158-f001:**
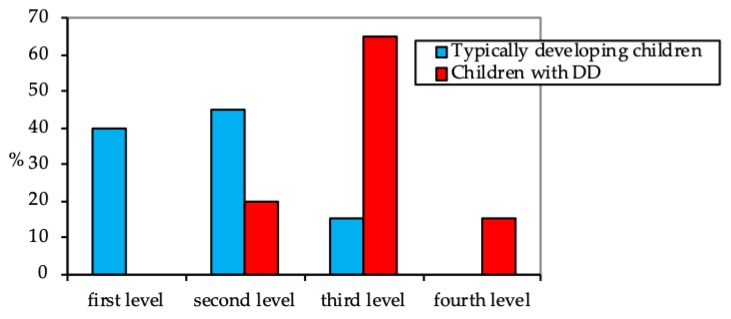
The dotting test results, %.

**Figure 2 behavsci-09-00158-f002:**
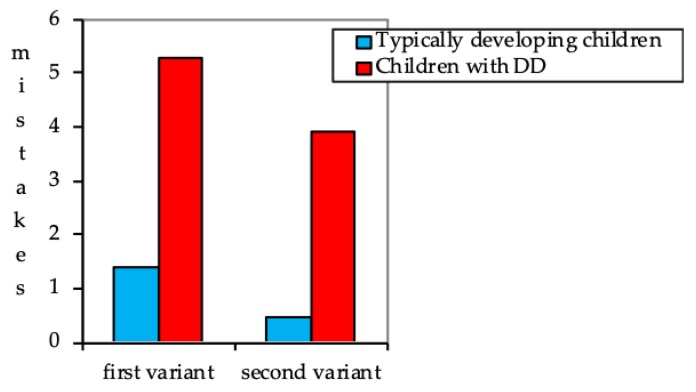
The Toulouse-Pieron attention test results (absolute values).

**Figure 3 behavsci-09-00158-f003:**
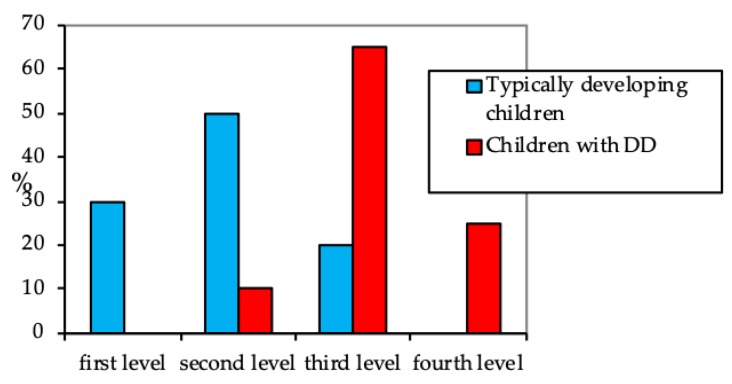
The visual pattern test results, %.

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
