# Peer review of "Experimental Research into Conscious Self-Regulation in First-Graders with Developmental Delay"

_behavsci, 2019, doi:10.3390/bs9120158_

Round 1
Reviewer 1 Report
This is important work and applies the concept of self-regulation to a younger population than is currently in the literature today. The author(s) are to be commended for this focus. I do think that there are some revisions to the existing manuscript to strengthen its relevance. They are summarized below:
Introduction/Background literature: This section should provide a strong rationale for the study, and with some changes, I believe it can be very strong. First, the information about children with DD is written as if the skill deficits related to self-regulation apply to all individuals with DD and that they are absolute. If they are absolute, then why attempt to teach these skills? Or further prove the fact that they exist? Instead, we know that DD is a broad term for a range of disabilities and skill deficits that children MAY experience. We have research in the field that has demonstrated that these skill deficits CAN be overcome and/or minimized; if the introduction focused on that approach, it would help strengthen the rationale for this research.
Methods: This section should be expanded to include more information about the instruments used in the study (why were they chosen? What is the validity/reliability of their use with this population of children?). Also, the steps of the implementation of this study should be detailed enough so that another researcher could replicate it. Without such detail, it is impossible to evaluate the strength of your findings or to know if the findings apply to other populations of children with and without DD.
Findings: Could not evaluate given the comments about the methodology section.
Discussion: The author(s) make the claim that their research findings support the development of a program that can be used to improve the self-regulation skills of children with DD, but again, because there was not enough information in the methods section, it is difficult to determine the strength of this multi-level approach. I recommend that this section include a table that shows the connections between the training steps (and what component skills of self-regulation they attempted to address), the findings related to that skill, and the components of the proposed intervention that includes that component.
I hope these comments are helpful to the author(s) and that they continue with this very important work.
Author Response
Response to Reviewer 1 Comments
Thank you for your comments and recommendations, they have been very helpful.
Point 1: Introduction/Background literature: This section should provide a strong rationale for the study, and with some changes, I believe it can be very strong. First, the information about children with DD is written as if the skill deficits related to self-regulation apply to all individuals with DD and that they are absolute. If they are absolute, then why attempt to teach these skills? Or further prove the fact that they exist? Instead, we know that DD is a broad term for a range of disabilities and skill deficits that children MAY experience. We have research in the field that has demonstrated that these skill deficits CAN be overcome and/or minimized; if the introduction focused on that approach, it would help strengthen the rationale for this research.
Response 1: We have added more information to introduction, it describes the range of difference within the category of children with DD. Research on a lack of self-regulating skills and capacities in children («the zone of proximal development» by Vygotsky) can contribute to overcoming and/or minimising developmental delays with targeted and differentiated approach.
Point 2: Methods: This section should be expanded to include more information about the instruments used in the study (why were they chosen? What is the validity/reliability of their use with this population of children?). Also, the steps of the implementation of this study should be detailed enough so that another researcher could replicate it. Without such detail, it is impossible to evaluate the strength of your findings or to know if the findings apply to other populations of children with and without DD.
Response 2: We significantly expanded the Method section. Now it has more detailed information on the participants. It also includes description of screening tools and protocol. To add, now there is more information about the indicators on which a qulitative analysis of the research results was based and according to which we came to the research conclusions.
Point 3: Findings: Could not evaluate given the comments about the methodology section.
Response 3: I hope that with the details added to the Method section the research results make more sense now.
Point 4: Discussion: The author(s) make the claim that their research findings support the development of a program that can be used to improve the self-regulation skills of children with DD, but again, because there was not enough information in the methods section, it is difficult to determine the strength of this multi-level approach. I recommend that this section include a table that shows the connections between the training steps (and what component skills of self-regulation they attempted to address), the findings related to that skill, and the components of the proposed intervention that includes that component.
Response 4: Thank you for making such valuable comments to this section. According to the recommendations of the reviewer 3, I'll make the self-regulation development programme a subject of a separate article in which I would follow your recommendation to include the table for programme reasoning.
Thank you again for your attention to my article and for very helpful comments! I appreciate it.
Reviewer 2 Report
Thank you for inviting me to review the paper on “Experimental Research into Conscious Self regulation in First-graders with Developmental Delay”. This is an ambitious study that found that the findings proved the children of 7-8 years with DD to have an underdeveloped level of conscious self-regulation of actions in cognitive activity and to differ from their typically developing peers in terms of conscious self-regulation level and skills. I have the following recommendations:
How were children with mental developmental delay (DD) being diagnosed? Were they assessed by pediatricians specialized in development or educational psychologists? Do they attend special schools? The authors need to provide more information. Under the method, the authors stated “60 children aged 7-8 years were involved in the study, with typically developing schoolchildren, and with DD” It should be sixty, not 60. The authors need to provide more information about the following tests under the method: “The practical methods used in the study are the «Dotting test», the «Visual pattern test», the «Toulouse-Pieron attention test». What do these tests measure? How were these tests administered? What was the time spent on each test? Under methods, the authors need to mention statistical analysis and how the data were analyzed and which tests were used? Under results, the following description mentioned by the author, “The «Dotting test» modified by Osnitskii A.K. [16] was applied to study self-regulation of actions and its capacity (determination and adherence to an activity program, acquiring and maintaining the learned actions needed to perform task-techniques, self-regulation skills transfer to new conditions) and the impact of different conditions on performance and efficacy of self-regulation in children with DD as opposed to their typically developing peers.” should be moved to methods. Please move other test descriptions to the method. Why Figure 1 and 3 are so similar? Could it be a mistake? The authors need to state limitations under discussion. There was no p-value for each figure. The authors need to explain why the sophisticated statistical analysis was not performed to compare results at different levels and between groups in Figure 1 to 3.Author Response
Response to Reviewer 2 Comments
Thank you for your comments and recommendations, they have been very helpful.
Point 1: How were children with mental developmental delay (DD) being diagnosed? Were they assessed by pediatricians specialized in development or educational psychologists? Do they attend special schools? The authors need to provide more information. Under the method, the authors stated “60 children aged 7-8 years were involved in the study, with typically developing schoolchildren, and with DD” It should be sixty, not 60.
Response 1: We have added information on the sample of children. Diagnostic screening of children with DD was performed by a panel of experts comprised of a developmental paediatrician, an educational psychologist and a special education teacher. Eight general schools in Moscow (Russia) were selected for the sample. All the children were first-graders, children with DD studied in these schools on terms of inclusive education.
Point 2: The authors need to provide more information about the following tests under the method: “The practical methods used in the study are the «Dotting test», the «Visual pattern test», the «Toulouse-Pieron attention test». What do these tests measure? How were these tests administered? What was the time spent on each test? Under methods, the authors need to mention statistical analysis and how the data were analyzed and which tests were used?
Response 2: We significantly expanded the Method section. Now it includes description of screening tools and protocols. It also has more information on the indicators on which a qualitative analysis of the research results was based and according to which we came to the research conclusions. Statistical analysis was applied only for considering the statistical significance between children with DD and their typically developing peers. In subsections 3.1, 3.2 and 3.3 you may find p-values.
Point 3: Under results, the following description mentioned by the author, “The «Dotting test» modified by Osnitskii A.K. [16] was applied to study self-regulation of actions and its capacity (determination and adherence to an activity program, acquiring and maintaining the learned actions needed to perform task-techniques, self-regulation skills transfer to new conditions) and the impact of different conditions on performance and efficacy of self-regulation in children with DD as opposed to their typically developing peers.” should be moved to methods.
Response 3: We added general information on the «Dotting test», the «Toulouse-Pieron attention test» and the «Visual pattern test» and their protocols in the Method section. From my perspective, the detailed description in the Results section could be helpful for a reader.
Point 4: Please move other test descriptions to the method. Why Figure 1 and 3 are so similar? Could it be a mistake?
Response 4: Truth be said, Figure 1 and Figure 3 do look similar. That has to do with the similar-looking results that different children achieved when they completed tasks.
Point 5: The authors need to state limitations under discussion. There was no p-value for each figure. The authors need to explain why the sophisticated statistical analysis was not performed to compare results at different levels and between groups in Figure 1 to 3.
Response 5: In the research the qualitative indicators were analysed in the first place. When possible, the quantitative analysis was carried out as well. In the Subsections 3.1-3.3 you may find p-values for each type of statictical analysis.
Thank you again for your attention to my article and for very helpful comments! I appreciate it.
Reviewer 3 Report
Thank you for the opportunity to review the manuscript titled, "Experimental Research into Conscious Self-regulation in First-graders with Developmental Delay." I appreciated the opportunity to read about your work. The subject of the article is interesting and relevant for the field.
The article begins with an organized introduction to the content of the article. Most of the references appeared to be Russian. It would be nice to include a few references from international literature to identify how the findings are similar to those around the world.
The English language and style are consisted throughout. There are some areas that would benefit from editing for spelling errors.
There are two areas in need of extensive revisions in the article. First, the methods section is lacking essential information to understand the study being described. To begin, the methods section needs a more detailed description of participants, setting, and measures used. Information is provided about the age and number of student participants, but no information is included about their race/ethnicity, socioeconomic status, or any other pertinent demographic information. The setting is not described in detail. For example, the region, country, city, etc. of the participants or where the data was collected. The participant and setting information is helpful for readers to understand the context of the research. Lastly, the names of the measures are listed but a detailed description is not provided for the readers. It is unclear to me what the student participants did, how they were grouped, how many were in a group, etc. All of the methodological information is needed to understand and contextualize the findings and remainder of the article.
It was difficult for me to determine the soundness of the findings and discussion without the methodological information described above. I would caution the author with using words live "proved" when describing the findings of the experimental research. I would also add in citations within the your discussion, such as to Vygotsky when talking about the zone of proximal development.
Lastly, I found it confusing when the author switched from a discussion about the findings to introducing an intervention program on pages 6 and 7 of the manuscript. I think it is important to identify next steps, but I did not think that explaining a program that was not used in the research fit in the manuscript for a full page of the article. I would recommend removing this section and writing a separate article about the intervention program.
Thank you again for the opportunity to review your manuscript.
Author Response
Response to Reviewer 3 Comments
Thank you for your comments and recommendations, they have been very helpful.
Point 1: The article begins with an organized introduction to the content of the article. Most of the references appeared to be Russian. It would be nice to include a few references from international literature to identify how the findings are similar to those around the world.
Response 1: We added 5 more references to international academic literature.
Point 2: The English language and style are consisted throughout. There are some areas that would benefit from editing for spelling errors.
Response 2: I've done my best to correct spelling errors.
Point 3: There are two areas in need of extensive revisions in the article. First, the methods section is lacking essential information to understand the study being described. To begin, the methods section needs a more detailed description of participants, setting, and measures used. Information is provided about the age and number of student participants, but no information is included about their race/ethnicity, socioeconomic status, or any other pertinent demographic information. The setting is not described in detail. For example, the region, country, city, etc. of the participants or where the data was collected. The participant and setting information is helpful for readers to understand the context of the research. Lastly, the names of the measures are listed but a detailed description is not provided for the readers. It is unclear to me what the student participants did, how they were grouped, how many were in a group, etc. All of the methodological information is needed to understand and contextualize the findings and remainder of the article.
Response 3: We significantly expanded the Method section. Now it includes more detailed information on the participants. There is screening tools and protocols description added. It also has more information on the indicators on which a qualitative analysis of the research results was based and according to which we came to the research conclusions.
Point 4: It was difficult for me to determine the soundness of the findings and discussion without the methodological information described above. I would caution the author with using words live "proved" when describing the findings of the experimental research. I would also add in citations within the your discussion, such as to Vygotsky when talking about the zone of proximal development.
Response 4: I hope that with the details added to the Method section the research results make more sense now. We also added quotations regarding self-regulation and the zone of proximal development by Vygotsky.
Point 5: Lastly, I found it confusing when the author switched from a discussion about the findings to introducing an intervention program on pages 6 and 7 of the manuscript. I think it is important to identify next steps, but I did not think that explaining a program that was not used in the research fit in the manuscript for a full page of the article. I would recommend removing this section and writing a separate article about the intervention program.
Response 5: I must thank you for making such a valuable comment. I've removed this section from the article and I’m going to write a separate article about the intervention programme as you recommended.
Thank you again for your attention to my article and for such valuable comments!
Round 2
Reviewer 1 Report
Good modifications.